# Successive Pandemic Waves with Different Virulent Strains and the Effects of Vaccination for SARS-CoV-2

**DOI:** 10.3390/vaccines10030343

**Published:** 2022-02-22

**Authors:** Alcides Castro e Silva, Américo Tristão Bernardes, Eduardo Augusto Gonçalves Barbosa, Igor Aparecido Santana das Chagas, Wesley Dáttilo, Alexandre Barbosa Reis, Sérvio Pontes Ribeiro

**Affiliations:** 1Laboratory of Complexity Science, Department of Physics, Universidade Federal de Ouro Preto, ICEB, St. Quatro, 786, Bauxita, Ouro Preto 35400-000, MG, Brazil; atb@ufop.edu.br; 2Centro Federal de Educação Tecnológica de Minas Gerais, Graduate Program in Mathematical and Computational Modeling, Ave. Amazonas, 7675, Nova Gameleira, Belo Horizonte 30510-000, MG, Brazil; eduardo.agbarbosa@pbh.gov.br; 3Graduate Program in Biological Sciences, NUPEB, Universidade Federal de Ouro Preto, St. Três, 408-462, Bauxita, Ouro Preto 35400-000, MG, Brazil; igor.chagas@aluno.ufop.edu.br; 4Instituto de Ecología AC, Red de Ecoetología, Carretera Antigua a Coatepec, 351, El Haya, Xalapa 91070, Veracruz, Mexico; wesley.dattilo@inecol.mx; 5Laboratory of Imunopatology, Department of Clinical Analysis, Universidade Federal de Ouro Preto, NUPEB, St. Três, 408-462, Bauxita, Ouro Preto 35400-000, MG, Brazil; alexreis@ufop.edu.br; 6Instituto Nacional de Ciência e Tecnologia em Doenças Tropicais (INCT-DT), Salvador 40000-000, BA, Brazil; 7Laboratory of Ecology of Diseases and Forests, Department of Biodiversity, Evolution and Environment, Universidade Federal de Ouro Preto, ICEB, St. Quatro, 786, Bauxita, Ouro Preto 35400-000, MG, Brazil; spribeiro@ufop.edu.br

**Keywords:** COVID-19, pandemic, vaccination, ABM-SIR model

## Abstract

One hundred years after the flu pandemic of 1918, the world faces an outbreak of a new severe acute respiratory syndrome, caused by a novel coronavirus. With a high transmissibility, the pandemic has spread worldwide, creating a scenario of devastation in many countries. By the middle of 2021, about 3% of the world population had been infected and more than 4 million people had died. Different from the H1N1 pandemic, which had a deadly wave and ceased, the new disease is maintained by successive waves, mainly produced by new virus variants and the small number of vaccinated people. In the present work, we create a version of the SIR model using the spatial localization of persons, their movements, and considering social isolation probabilities. We discuss the effects of virus variants, and the role of vaccination rate in the pandemic dynamics. We show that, unless a global vaccination is implemented, we will have continuous waves of infections.

## 1. Introduction

At the end of 2019, the world received news about a novel disease that started in Wuhan, China. This illness, called COVID-19, is caused by a SARS class virus named SARS-CoV-2. Due to its high transmission capability, the disease rapidly reached all countries around the globe, mainly through airports networks [1,2,3], and, on 11 March 2020, the World Health Organization (WHO) declared it a pandemic. As of September 2021, the Johns Hopkins COVID-19 dashboard [4] showed more than 200 million cases and more than 4 million deaths globally. Such a number shows how fast the SARS-CoV-2 can spread through an entire population if no coordinated actions to prevent and reduce infections are in place.

Any responsible sanitary policy adopted to slow down the progress of COVID-19 pandemic should make use of the following three strategies:Social distancing: the obvious way to reduce susceptible-infected interaction and subsequent contagion;Mask wearing and hygiene: this was implemented once it became known that transmission is mainly through respiratory droplets of infected patients and contact with surfaces infected by aerosol;Vaccines: a correct vaccination program can decrease overall transmission and the intensity of the disease symptoms among those infected and vaccinated, reducing the public health collapse risk and the mortality rates, as susceptible but vaccinated people become asymptomatic. Still, the virus will circulate, and the lack of a proper vaccination will create outbreaks due to contact between an increasing number of “asymptomatic” people with susceptible people. As [5] demonstrated, the existence of transient collective immunity may prolong an epidemic, and a bad vaccine scheme may exacerbate this pattern.

For the specific case of COVID-19 vaccination, one of the subjects of the present work is that there are many factors that must be considered for a suitable immunization policy. Among all of them, this work focuses mainly on two aspects: how the virus is evolving into new variants of concern and reinfection.

Every time SARS-CoV-2 infects a susceptible person, it starts to make copies of itself replicating its RNA [6,7]. As this process is not 100% error proof, some changes can be introduced, and different copies can be created. These changes or mutations in RNA can lead to different scenarios: it can be an evolutionary dead end and kill the virus, it can be an irrelevant and not noticeable change, or it can bring some advantages, for example, better response to the immune system or a better enhanced ability to invade the cells [8]. Even more rarely, whole clusters of mutations can be acquired by the virus during a single infection. When a virus with a single or clusters of mutations is capable of an epidemiologically significant spread through populations, they are named “Variants of Concern” or VOC. According to US Center of Disease Control and Prevention (CDC) [9], a variant becomes a VOC *when there is evidence of an increase in transmissibility, or in lethality and severity of the disease, significant reduction in neutralization by antibodies generated during previous infection or vaccination, reduced effectiveness of treatments or vaccines, or diagnostic detection failures.*

Although there are thousands of different genetic lineages of SARS-CoV-2, there are few Variants of Concern [10]. Both the variant β (former B1.351), which was first detected in South Africa, and variant γ (former P.1), which was first seen in Brazil and Japan, contain mutations that appear to weaken the ability of antibodies to neutralize the virus by binding to it, which would normally prevent it from infecting cells [11]. Variant α (former B1.1.7), detected in the UK and reported in 93 other countries, and variant δ (former B.1.617.2), from India, show less of an ability to escape from antibodies, but they have gained mutations that allows them to reproduce faster, rapidly increasing the viral load in an infected person, with consequences for the transmissibility than the original version of the virus [12,13]. The latest Omicron variant has an extremely high transmissibility, even higher than δ, but it is less likely to infect the lungs [14,15].

Apart from VOCs, there are still the “Variants of Interest” or VOIs (variants ε, η, ι, κ, and ζ) and “Variants of High Consequence” (there are no SARS-CoV-2 variants that rise to the level of high consequence until now) [16]. The CDC definition of a VOI is a variant with specific genetic markers that have been associated with changes to receptor binding, reduced neutralization by antibodies generated against previous infection or vaccination, reduced efficacy of treatments, potential diagnostic impact, or predicted increase in transmissibility or disease severity. On the other hand, a variant has high consequences when there is clear evidence that prevention measures or medical countermeasures (MCMs) have significantly reduced effectiveness relative to previously circulating variants.

The second factor that can impact an immunization policy is the reinfection caused by the loss of immunity. Some works have shown that immunity with greater memory is acquired by infected people who developed severe symptoms, recovered, and were vaccinated with at least one dose. However, even in these cases, immunity is not permanent, requiring a new dose of vaccine [17,18]. The reinfection was assumed as rare just before the spreading of the new variants of concern in early 2021 [19]. However, risks may rise if the pandemic is not controlled and the virus is left to evolve freely [20,21,22]. Reduced neutralization of the Delta variant in comparison to the ancestral Wuhan-related strain was already observed [23], and a complex relation between different variants is also possible. For instance, it was found that people infected with Beta variant are more susceptible to reinfection by Delta variant [23]. Hence, there is room for the evolution of new variants that could escape vaccination more aggressively, especially if the vaccination scheme continues to follow a heterogeneous pattern, leaving the most vulnerable populations exposed for longer [24].

In this work, we developed an epidemiological model where events such as the appearance of new variants and reinfection are taken into account. Our results point to an optimal vaccine frequency that should be conducted in a given epidemiological setting.

Mathematical models for the evolution of epidemics have been proposed in the last century; one with the concept of compartmental models was introduced by Ross (1916), followed by the most known and referenced model, proposed by Kermack and McKendrick (1927) [25]: the SIR or “susceptible-infected-removed” model (nowadays, the term recovered is also used). Many incremental evolutions of the original models have been studied, by considering many other “compartments”, such as non-asymptomatic, hospitalized, and other situations.

While in the ordinary versions of SIR and its descendants, the population is simulated by densities, which assumes an infinite population and does not capture effects of finite or even small communities, in the present work, an Agent-Based Model [26] version of the SIR model is introduced. The population is represented by individuals which can interact, but the health conditions are defined by a few states. This allows us to better understand the pandemic dynamics.

## 2. Materials and Methods

### The Model

In the present paper, we simulate a version of the SIR model [25] in a city with a population N (t) that may vary over time. Many variants of SIR model were used to simulate different scenarios of the SARS-CoV-2 pandemic [27,28,29]. However, to have a better understanding of how cities’ structures and citizen dynamics affect the spreading of a transmissible disease such as SARS-CoV-2, we chose to develop an ABM-SIR (Agent-Based Model) [30,31,32,33] of a “city” in which citizens start to become infected.

The city represents a geographically limited region in which only the arrival and departure of visitors and the deaths of its inhabitants can change its population. As in the previous versions of the SIR model, a *S*_i_ variable defines the health state of each individual: susceptible (*S*_i_ = 0), infected (*S*_i_ = 1), and recovered or immunized (*S*_i_ = 2). In this work, we have included a fourth state: dead (*S*_i_ = 3). Factors such as age, sex, or race are not considered.

In the simulated city, the residents live in houses, and they can move to public establishments (e.g., such as malls, stadiums, stores, restaurants, etc.). In some simulations, people can move to other people’s houses. The day starts with all the residents in their homes, to which they will be linked throughout the simulation. That is, if they leave for another home or any establishment, at the end of the day, they will return to their own home. Each person has a probability of movement *P_mov_*, and this probability defines the social isolation mobility.

There is no natural movement, and to go from one location to the next takes zero time. The agents disappear from a place and reappear in another. We considered that the time of permanence in public or private places is longer than the travel time. On the other hand, for big cities, one can suppose that the time they stay in a mode of public transport, if prolonged, may also be considered as staying in a small public place with the same infection conditions.

Every public area has a carrying capacity of *K*, so that Σ *K*_i_ = *N*_0_, that is, the city allows all of its residents to leave at the same time. There are several public areas with different *K*_i_ = {10, 000, 1, 000, 100, 50}, so that the sum of the capacity of each area of the same type is equal to 0.25 *N*_0_. The number of places of a specific kind i is the total number of individuals they can support divided by its *K*_i_. Thus, the number of sites that support fewer people is higher than those which support more people: in other words, there are more smaller stores than big stadiums. When individuals can move to other people’s houses, we define a carrying capacity of 12 individuals.

The time scale of our simulations is one day. For each day, individuals are chosen at random, among the number of alive people on that day. A resident can either not be drawn or could be drawn more than once. After this, we test if they will move, with the probability given by *P*_mov_. Each individual may visit other places a maximum of three times during the day. If a person is going to move, the places to which they will go are chosen at random: houses, small shops, or large stores. Its maximum capacity gives the likelihood of going to a location: they are more likely to go to a large store than to someone else’s home. If a selected place reaches its capacity, a new site is drawn until the person moves, ensuring that whoever was chosen to move will make a move. COVID-19 has a high transmissibility during the pre-illness period, and the model assumed seven days of transmission before causing any disease, which was estimated by the pandemic for previous variants and the Delta variant. The mobility defined in the model emphasizes the transmissibility by pre-symptomatic and asymptomatic individuals, as no constraint on the movement of an infected person is imposed until someone dies [34,35,36].

The person arriving at a new location may be in one of three states: susceptible, infected, or immunized. If the person is vulnerable, it is verified if there is someone infected at that location. If not, the person stays there until they move again or until they return home at the end of the day. However, if there is an infected person(s) at that location, first, we calculate the probability of contact with the infected person, given by:(1)$$P_{contact}=\frac{N_{infected}}{N_{max}}$$
where *N*_infected_ is the total number of infected persons at that place (house or mall) and *N*_max_ is the maximum capacity of that place.

Two steps define the infection of a susceptible person: first, we calculate if they come into contact with a contaminated person. If so, we test if they will contract the virus, as contact does not imply infection. The probability of infection is β. In the case of COVID-19, β is estimated between [0.2, 0.3]. In the cases simulated in this work, we use a value of β = 0.2 for a “normal” variant and we assume β > 0.2 for more transmissible variants [37].

The probability of becoming infected with one of the variants present in that location is proportional to how often that variant is in that location, weighted by the transmissibility of the strain. The higher the *n*_i_ × β_i_, the higher the probability of being infected, where *n*_i_ is the total number of individuals infected by the strain β_i_ at that location. The infected individual then becomes a potential transmitter of the disease.

If an individual who arrives at a place is infected or immunized (*S*_i_ = 1 or 2), nothing happens to those who are already at that place. The arrival of an infected person will create the conditions for those who arrive later to be infected. Hence, there is an asymmetry in the model. If an infected person arrives at a place, we do not test if they will infect the susceptible ones already there. This kind of procedure corresponds to a sequential order in the contact/contagion process. It is understood that for huge populations and many days of simulation, the results will not be different from those here reported.

At the end of the day, after performing some movements, all individuals return to their homes. Of course, if they have not moved, nothing is tested. However, for those who return to their home, the same set of procedures to check for infection is adopted: If the person returning home is susceptible, and someone infected is in the house, the same steps described above are carried out for contact and contagion.

The dynamics can be summarized as follows: To each individual *i* is assigned a variable *S*_i_ that can be in four states: susceptible (0); infected (1); recovered or immunized (2); and dead (3). The entire population starts the simulation as susceptible: *S*_i_ = 0. A single person among the residents is infected at *t* = 0 with the less lethal variant β = 0.2. The dynamics is given by the process shown in Figure 1.

The figure represents the possible changes of state. A susceptible individual *S*_i_ = 0 can become infected *S*_i_ = 1 with the probability given by *P*_contact_ × β_i_. β_i_ is the transmission rate of an individual which contaminates a susceptible individual, as described above. An infected individual *S*_i_ = 1 can die *S*_i_ = 3 if their time of infection is greater than 7 days and with probability of death given by *P*_death_. In case of contamination with the most lethal variant β + δβ, the value of *P*_death_ is increased by δ_death_. This antagonism creates a tension between the strain’s lethality and the individual’s probability of death. That is, hosts with a more lethal variant are more likely to die.

The state of infection in an individual remains for a maximum of 1/γ = 14 days. It is assumed that the potential for infecting another person does not change during this period. It depends only on the β value, which does not change for a contaminated individual.

We have also simulated the case of incubation. In this case, the susceptible individuals become infected after five days of incubation.

After 1/γ days, the infected individual, *S*_i_ = 1, either recovers or is immunized, *S*_i_ = 2. After *T* days, the immunized individual, *S*_i_ = 2, returns to the susceptible condition, *S*_i_ = 0. In the case of a vaccination program in that community, a susceptible or infected individual, either *S*_i_ = 0 or *S*_i_ = 1, can be immunized with probability *P*_vac_. This probability is related to the vaccination rate, which is the percentage of the total population that is vaccinated every day after the campaign started. The status of a vaccinated individual becomes *S*_i_ = 2. Similar to the recovered one, the immunization protects an individual for *T* days. During this period, they cannot be infected with any variant of the new coronavirus [38,39].

In our simulations, vaccination starts from the 300th day. This date roughly corresponds to the beginning of vaccination campaigns in several countries: between December/2020 and January/2021, considering the detection of the pandemic as time zero (around February or March 2020). There is no need to use two doses or intervals between doses of the vaccine to gain immunity. In this simplified version of the model, immunity is acquired at the time of vaccination. Although it is possible that vaccinated individuals can infect and transmit the virus, it happens more rarely or with lower intensity than with infected non-vaccinated individuals; thus, for sake of simplicity, in our model, vaccinated individuals did not get infected [40]. Due to its structure and complexity, we used the Brazilian vaccination numbers as reference. We performed simulations with two vaccination rates. Some simulations used a vaccination probability of *P*_vac_ = 1/200, representing the typical value of vaccination campaigns in Brazil, when close to 1 million people are vaccinated per day. Brazil has a public health system formed by about 40 thousand health centers, belonging to three levels of administration, but forming a cooperative network. This system has vast experience on vaccination campaigns and can easily reach a rate of 1 million vaccinations per day, roughly 1/200. The second value we have tested is a rate of 1/1000, which represents the values that we have observed for the vaccination against COVID-19 in the first two months of vaccination, close to 200 to 300 thousand people per day. From 23 January 2021 to 29 March 2021, 14 million people received the first dose of one of the two available vaccines. In Brazil, the vaccines that were used were those that required two doses. The vaccination rate increased in April but then oscillated, since Brazil had no plan of vaccine acquisition. In the USA, the vaccination rate reached 0.006 of the whole population in March. By the beginning of March, Israel had already vaccinated practically its entire population of around 9 million people.

Our model assumes that visitors arrive and leave the city every day. The number of visitors coming and leaving each day is around 1/1000 of the total residents. These visitors have a probability of movement equal to 1. This means that a visitor will go to three places (houses, small shops, or large stores). The likelihood of a visitor being infected is 0.1%. In principle, if infected, they will have the virus variant β = 0.2, but some can have a more transmissible variant. We have implemented a version in which visitors are 1% of the population. Of the visitors, 3% may have the new, more transmissible type of virus.

If that visitor is susceptible, they may become infected during the day and transmit the infection. However, as they only spend one day in the city, they will not be subject to death, nor will they be able to recover. They will also not be vaccinated.

In this model, there are no births. There is no increase in the number of inhabitants, except the increase caused by visitors, which represents a zero change in the population, as visitors only arrive and leave. The only factor that can change the resident population is death caused by disease. We study the effects of a new variant of the virus, which enters via visitors, studying the competition dynamics between different strains. Moreover, we also want to study the effects of vaccination, not considering other variables. A local variant is not supposed to mutate.

We show the results with two versions of the model: in the first one, we simulate a city with houses and small and big stores, where the visitors may carry just one virus variant with higher transmissibility. Residents can move to other residents’ houses. In the second version, the towns have homes and just one type of store, and the visitors carry many virus variants (described below).

We studied the effects of vaccination only in the first version of the model. The central aspect of the first version is to look at the spread of a higher transmissibility strain with vaccination. In addition, we used real COVID-19 cases and vaccination strategies from two populations in similar countries, i.e., Portugal and Israel, to validate our mathematically based conclusions. The second version aims to study the competition between different strains, and thus, the natural evolution of the pandemic without vaccination. In both cases, we consider that variants with different transmissibility also have different lethality.

In this work, as usual in computer simulations, we assumed some simplifications and assumptions in order to optimize the model, and to capture the most significant possibilities according to some objectives. One of these simplifications is the absence of traffic and commuting, i.e., the time required for displacement. The introduction of a transportation network could improve the model, and is likely to bring new features to the study. Another limitation of our work is the population structure. In our “city” model, the population does not have any age or gender structure. For COVID-19, it is known that the response to infection can vary depending on the infected age; thus, the use of an age-structure AMB-SIR model also could bring more realism to the simulation. Finally, our work assumes that all variants are affected equally by vaccinations, which does not reflect the real-life situation. Thus, adding different responses to the vaccine–variant interaction could strongly improve the model. Similar to other models, as we have discussed, we tried to shed some light on the main effects of the dynamics. On the other hand, the assumptions we made in order to allow a feasible but simplified model to run expectations of the effects caused by an effective vaccination program allowed us to test the difference between having or not having an effective vaccination program in an open-to-migration city during an evolving pandemic. Thus, our model considered a successional dynamic of variants with distinct life traits. Even the simplifications we assumed did not undermine the effectiveness of vaccines, considering we built up a more vulnerable society (where everybody was equally susceptible) combined with realistic combinations of lethal-transmissible variants.

## 3. Results

### 3.1. Model with Two Strains

The first set of results were obtained in simulations of cities with two populations: N ∼875 thousand residents and N ∼8.75 million residents. In the first case, they live in 250 thousand households. When allocating people, it was decided that a house had 3 or 4 residents; thus, there were 3.5 residents on average per household [41]. These residents can move to 20,000 stores with a maximum capacity of 50 people or 2000 establishments which support 500 people. The houses, small shops and big shops can accommodate all citizens, which means that there is enough space for all people to move to any location.

In the simulations described below, values of *P*_mov_ = 0.6, 0.5, or 0.3 were adopted, meaning an average isolation of 40%, 50%, and 70%. In the case of Brazil, the average isolation, measured while individuals stay in their homes for a day, has fluctuated between 30% to 40%, and on weekends, it can reach 50%. It increased a little in February and March 2021 due to the imposition of new measures of restriction and operation of the establishments. At the beginning of the pandemic, in March 2020, there were higher values in Brazil, in the range of 60%.

For each day, individuals are chosen at random, as described above. It was verified if they were going to leave the house. If so, we drew the total number of moves they would make: 1 up to 3. Then, we randomly chose the places for visitation: other houses or small or large stores. Then, they left the house and went to each establishment. Each person executed their movements, and after completing them, stayed at the last site until they returned home at the end of the day.

In all the simulations discussed below, 10% of the population had a probability of movement of *P*_mov_ = 0.1, representing the people who were most often at home. As described above, visitors would come and go. We assumed a proportion of visitors of 0.1% of the total initial population per day. The people who arrived and left remained in the city for only one day. A ratio of 1/1000 may be contaminated. The contamination strain had β = 0.2. In total, 1/90 of the contaminated individuals had a more lethal variant, with β = 0.25. Visitors had a probability of movement equal to 1.0 and always visited three locations (drawn at random, as described above).

In the results shown below, *P*_contact_ is given by Equation (1), β = 0.2, δβ = 0.05, *P*_death_ = 0.01, δ_death_ = 0.004, 1/γ = 14, and T = 120 days.

On the simulation’s 300th day, a vaccination campaign could start. Figure 2 represents the evolution of a population with 40% of social isolation, *P*_mov_ = 0.6, without vaccination. The values represent the densities relative to the initial number of residents, in this case, 874,975 people. There is an oscillating evolution of the numbers caused by the ongoing entry of infected individuals: the visitors. The curve called β_av_ represents the average value of β taken among all individuals with *S* = 1. It is observed that in the peaks of infection, the most lethal variant tends to dominate. The decrease in the infected population is typical of the SIR model, but the permanence of the most lethal variant is an important characteristic. The number of individuals who died is in the order of 10% of the original population. Certainly, this is a very high number when compared to real data. This model and the simulations do not intend to project expected values, but they represent the dynamics of disseminating the most lethal variant qualitatively.

Appendix A shows the percentage of people dying each day (blue line) and the percentage of people who died due to the most lethal variant (red line). The results were obtained from the simulation described above. One can observe that the number of people who die each day increases in the periods of infection. However, the quantity of those who die from the new variant rises more rapidly, becoming ∼50% at the end of the simulation.

Change in this kind of dramatic evolution is only achieved when social isolation is about 70%, without vaccination. Appendix A shows a simulation obtained with the previous parameters, but now with *P*_mov_ = 0.3. In this case, the susceptible population remains basically between 0.95 and 1, the density to the original number of residents. Infected people cannot contaminate significant fractions of the population, and the infection stops.

Figure 3 shows the result obtained with vaccination at a rate of 1/1000, with a probability of movement of 60% corresponding to a social isolation of 40%, the peak value currently in Brazil. The numbers of infected and the progression of the pandemic do not stop. The numbers of deaths and the spread of the most lethal variant follow the patterns observed previously. This happens because the rate of vaccination, that is, the transition from susceptible and infected to immunized, is low compared to the rate of change from vaccinated to susceptible. Note that the time for this transition is *T* = 120 days, meaning four months on our time scale.

When the immunization rate assumes a value of 1/200, the picture changes radically, as shown in Figure 4. Vaccination started on day 300 of the simulation. One can observe a second wave of infections because the virus is widespread in the population. However, the number of immunized people increases rapidly, which is a blocking factor for the spread of the pandemic. One can also observe that the most lethal variant is contained, as the value of β_av_ is closer to 0.2. It is important to remember that the rate of contaminated visitors is the same in the three cases discussed so far. The number of deaths also stabilizes with the blockade caused by vaccination.

Appendix A reproduces the relationship between daily deaths and the presence of the most lethal variant. This variant is still present in the second wave but with a smaller percentage than in Appendix A. Later, this variant becomes marginal. It is necessary to remember that this most lethal variant enters the population from the visitors, and therefore, in this model, this variant will always be present.

In summary, we show that there is a dynamic relating the number of infected people to isolation and vaccination rates. Without vaccination, we observe that only with an isolation rate of 70% the pandemic is stopped. On the other hand, when the vaccination rate is 1/1000, a small value, it is observed that there is no significant impact on infection dynamics. Only with a higher vaccination rate, as in the cases of Chile or Portugal, can the pandemic be stopped, even with lower isolation rates.

To better understand the influence of the city size, the number of individuals, as well as the period of immunization after vaccination, we performed simulations with *N* = 8.75 million individuals and T = 180 days, meaning that a recovered or immunized stays in the *S* = 2 state for six months.

All the parameters are the same as those used in the simulations described above: *P*_mov_ = 0.6; β = 0.2 for residents; number of visitors, 1/1000; contaminated visitors, 1/1000; contaminated visitors with the highest transmissibility rate, 1/90. Vaccination, when it occurs, starts on the 300th day.

Qualitatively, the behavior observed in the curves is the same as discussed above for populations that are ten times smaller. Notice that the change in the immunization period *T* does not change the evolution of successive waves for the cases without vaccination or with a low vaccination rate. 

Figure 5 shows the evolution of cases and vaccination of two countries with approximately the same population, Israel and Portugal [42]. It is clear how the difference in the vaccination policy adopted in those two countries has shaped the cases curve. Both countries started their vaccination around the same period (12/2020). However, the rate of vaccination was completely different. Israel started at a high rate, but it faded around three months later, while Portugal kept an increasing rate. The impact in the cases can be seen in the last peak caused by the Delta variant in the last months of 2021, which was much stronger in Israel than Portugal. The huge increase of cases on the right of both plots (early 2022) are due to the spread of Omicron, which is not affected by vaccination. 

### 3.2. Model with Many Strains

We have implemented a second version of the model where residents can move only to public places. We have considered a more significant number of visitors who can bring different variants of the virus. In this version, for each instant of time (day), residents leave their houses with a probability that defines social isolation. They stay at home or move to public areas, making one or two movements per day. We simulated the initial isolation of 45% as default. In this second model, the flux of visitors is 1% of the total population, but infected visitors are 3% of the total number of visitors. All the other parameters are the same as the original model.

The objective of the second version of our model is to simulate how different strains compete among themselves in an epidemic scenario. This is achieved in the following way: the simulation starts with just one type of variant (called variant 1); after 5% of the population gets infected, four other variants (2, 3, 4, and 5) are introduced via infected visitors. These variants have different contagion and mortality probabilities, as can be seen in Table 1. It shows that, compared with the original strain, we have a combination of 70% and 50% more and less contagious and lethal variants, respectively.

Appendix A shows the evolution in a simulated city with a population of 1 million, with 45% of social distancing, and a more contagious variant introduced via visitors. The result shows the waves of infection caused mostly by loss of immunity after 120 days in a similar way to the results of the first model without vaccination.

The competition among different variants can be better viewed in Figure 6. This plot shows, for the same parameters of the previous plot, how the different strains are present in the population over time. It can be seen how the original strain causes a peak of ∼45% of people infected in a first wave around day 80. However, once the four new variants are introduced in the system, the following waves show that the original variant is rapidly replaced by the two most transmissible ones (variants 2 and 3 of Table 1). Since the second wave (starting between days 200 and 300), the simulation shows a dominance of variant 3 (more transmissible, less lethal), followed by variant 2 (more transmissible, more lethal), until finally, the original variant is barely present. This plot shows two important results: (A) the new variants with less transmissibility than the original strain (variants 4 and 5) are not able to invade the city, while the more transmissible variants substituted the original one from the second wave on; (B) interestingly, the lesser lethal variant overcame the more lethal one in all the waves, which is expected for a long-term evolution of an emergent disease if the lack of host illness favors transmission, and thus, loss of virulence [43,44]. Due to the full shifting of the original strain by new variants, the β-infection rate was permanently higher after the second wave than it was at the beginning of the pandemics, as shown by the average transmission in Figure 6.

## 4. Discussion 

The majority of papers modeling vaccination strategies are likely to fit in three categories: (1) extrapolation or estimation using real epidemiological data, for example [45], (2) compartmental models (SIR, SEIR, etc.), based on differential equations and epidemiological quantities defined as density of states (susceptible, infected, recovered, etc.), as in [46,47,48], and (3) Agent-Based Models (ABM) [32,49], which can be defined as an epidemic dynamic that takes place in a discrete way, with host/people as individualized entities of action, called “agents”. Models in family (2) have variables defined in a continuum and, thus, there is no “person” or “agent”, while in (3), ABM models are more versatile to describe a population where a set of characteristics must be considered, for example, age structure, topological distribution, and all kinds of spatial and temporal data. 

Models based on estimates taken from real data can be useful if high-quality data are available, with good extrapolations as a response to a very specific scenario. However, such models fail in exploring multiple parameters and “what if” questions. Compartmental models compose the majority of the publications and are very powerful tools. Although SIR models can deliver similar results as ABM, they are not as versatile as ABM models. As the ABM dynamic does not rely in differential equations, any change in the model can be done in a punctual way, only changing the agent parameters. Generally speaking, compartmental models are used in more generalist problems, while ABM appear to model real places.

Our model is a hybrid ABM-SIR, meaning it has a base structure of agents while the infection logic follows the SIR model. Using agents, our model was capable to sustain a population that moves in an independent way, while this same population could have citizens in different epidemiological states, stages, and places. This construct made it possible, with few modifications, to find out the optimum vaccination rate, as seen in Figure 4, and the competition among many variants in Figure 6. Our group proposed this novel hybrid model to compare the Spanish Flu with COVID-19 dynamics in the Brazilian city of Manaus [20].

After some waves of infections of COVID-19, the world is facing a new challenge. By the end of January 2022, 61% of the world population received at least one dose of the COVID-19 vaccine out of a total of 10.06 billion doses administered globally [50]. However, only 10% of people from low-income countries were vaccinated with one dose at that date. Even in largely vaccinated countries with severe inequalities, such as Brazil, there were areas with low vaccine coverage [51]. Additionally, vaccine hesitance and rejection spread in the developed world due to misinformation. Consequently, countries with a larger availability of vaccines and an early vaccinal program, such as the US and Germany, had a smaller proportion of overall vaccinated people by end of January 2022 than Brazil and Argentina, for instance, where anti-vax movements are irrelevant. Brazil and France vaccinated proportionally the same at this date, but France presented a more significant resistance against it, as in contrast to Brazil, the absolute availability of vaccines in France was higher for both adults and children [50]. Although it is a global problem, vaccine refusal has been properly studied mostly in the US [52,53,54,55]. Both vaccine inequality and hesitancy are likely to cause the spread of new variants [56]. Vaccination slowed down and this may cause the present situation [57,58]. In Israel, a new wave has been observed. However, since they have a high vaccination rate, the number of deaths is smaller than in the previous wave [59]. Brazil presents a different situation. The country started vaccination at a low rate, due to the delay to acquire vaccines from the federal government [60,61]. However, there was a strong engagement of the population and, after public pressure and the action of political sectors, the rates of vaccination increased in the second semester of 2021. The number of contaminated and dead people plateaued during the first months of 2021, but declined due to the increase in the vaccination rate. Different scenarios can be observed in different countries. However, they are a product of the spread of new variants and the vaccination of the population.

In this paper, we simulated the competition between virus strains, the role of the vaccination rate, and social isolation, which are believed as one of the main aspects to detain the virus’s circulation. In many countries, such as Brazil, it is very complicated to maintain social isolation for long periods. For instance, we observed, in parts of the population, stress among children out of schools, increase in domestic violence, and difficulties for individuals to obtain a basic income to sustain their families [62,63,64]. Poverty increased in many countries. In the case of Brazil, social isolation never reached adequate or recommended levels, which was responsible for the long plateau of infections and deaths.

Thus, we show that the only way to stop the circulation of the virus, or at least to diminish the contamination rates, is to increase the vaccination rates despite the current vaccination hesitancy and challenges to mass vaccination. Low vaccination rates allow the circulation of the many variants, and we observed a cyclic problem. The situation gains a much more dramatic feature with the existence of new variants with a higher transmissibility.

The many different variants will compete and those with higher transmissibility will win this competition, even in cases of combined higher lethality. This picture was observed anywhere the δ variant appeared. This variant rapidly infected people, even those already vaccinated, and caused a higher increase in contamination [65]. Thus, it brought humanity back to the beginning of the pandemic, struggling to flatten the infection curve, as even non-lethal new variants can overload health systems.

In a world where “vaccine nationalism” (i.e., governments supply the population of their countries with vaccines ahead of them becoming available to other countries) prevailed once more, the pandemic might find its way towards a natural evolution in those less vaccinated corners [66]. Our mathematical model was built considering a continuous immunization, with a single dose, and with a regular immunization rate in a scenario with no vaccine shortage. In addition, our mathematical model considered that vaccination was able to generate a four-month temporary immunity with only one dose, so immunization and booster doses every six months were not included for the calculations that we propose in this work. However, it is important to consider that the emergence of the Delta and now Omicron variant further reinforces the importance of access to COVID-19 vaccines, globally and equitably, for the health of all [67]. Constrained vaccine supply has driven opportunities for SARS-CoV-2 to mutate and be more infectious. 

The emergence of Omicron has emphasized that further delay in widely delivering at least the first two doses is fraught with peril for all [68]. On the other hand, our model reinforces the need to plan more strategic and specific vaccination schedules, to review current vaccines, and to design vaccines with the ability to prevent infection. The current vaccines have been overtaken by variants of concern and strains that can escape the protection given by the vaccine, despite maintaining protection against hospitalization and deaths. Nonetheless, we agree with the recommendation that in countries with a high prevalence of previous infections and a low proportion of over-60-year-old people, prioritizing delivering the first dose will have the greatest effect on preventing severe COVID-19 cases due to high vulnerability [69,70]. Conversely, in countries with a low prevalence of previous infections and a high proportion of old people, protection against severe disease in adults requires at least two doses, as well as booster doses in people who are severely immunocompromised or older than 60 years. Evidence suggests that although booster doses for all adults might prevent severe diseases, it could also compromise timely global availability of first doses [36].

Vaccination, more than natural infection, in addition to inducing the production of neutralizing antibodies, establishes a cellular immune response by activating memory T cells, which leads to a favorable clinical outcome and considerably reduces the number of deaths [71,72]. The most recent data have clearly shown that 70% of ICU admissions in European countries, the USA, and even in Latin America (Brazil) are people who have either not been vaccinated or who have not had two or three complete vaccine doses [73]. Finally, vaccination changes the dynamics, as it reduces the time of infection in vaccinated individuals and, consequently, interferes with the dynamics of the virus in the human body, preventing serious, long-term infections and, consequently, reducing the risks of new variants [74]. The existence of countries with low vaccination coverage is worrying, as in some European countries and several others African countries [75]. Particularly for some African countries, there is still a concern about the number of HIV+ and, thus, immune suppressed people without access to treatment, which may become sources for the emergence of new variants [76,77].

The Omicron strain, detected by the end of 2021, in South Africa, might fit the most consistent theoretical prediction: evolution of virulence loss [78]. This strain is amazingly contagious, spreading in a substantially faster rate than even the Delta strain, but which, as first evidence suggest, causes mainly mild symptoms, and may open the path for wide non-lethal dissemination [79]. Such a variant, if it indeed appears with these specific traits, might, and already has, overcome numerically any other, as Delta did before. However, the theoretical prediction of an emergent disease may not help lower the human and social costs. For instance, what level of hospitalization will be required with a too fast spreading strain, even if it is not lethal? Moreover, what sort of sequelae will it leave, increasing the public health cost of so-called long COVID syndrome?

## 5. Conclusions

The sudden disappearance of historic viral pandemic events might all have followed the path of natural selection favoring transmissibility against lethality, but there is a catch with SARS-CoV-2. Because transmissibility is high before symptoms, the positive selection based on losing lethality is weaker than it was, for instance, for Spanish Flu [20]. Even though natural loss of lethality may happen to a strain rapidly spreading worldwide, other strains may also appear in many places with non-fully vaccinated individuals and in places which are less internationally connected. There is, likewise, no impediment for new mutants which transmit fast and are still lethal. Hence, outbreaks of more virulent strains may keep surging for decades, according to our previous and present models [2,20]. The most relevant fact is that a planet of nearly eight billion people should not allow the luxury of leaving a pandemic to evolve widely.

The main purpose of this paper is to shed some light on the main aspects of the actual dynamics of the pandemic. A global governance is needed to deal with the immunization process, as the pandemic dynamics exhibits successive peaks with a distance of about 4 months between them. Without global control, new variants will continue to appear and when the infection is partially controlled in one country or region, but increases in other parts, a dramatic cyclic wheel of death may prevail.

## Figures and Tables

**Figure 1 vaccines-10-00343-f001:**
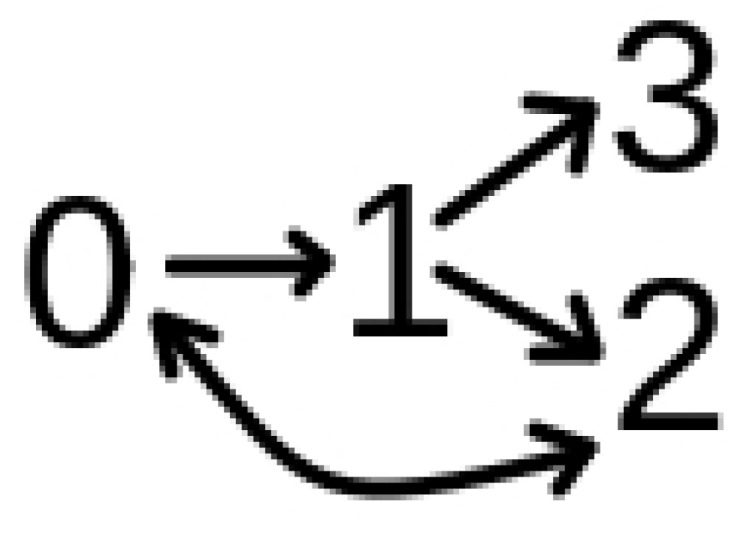
Schematic representation of transitions between states in our model. Probabilities are discussed in the text.

**Figure 2 vaccines-10-00343-f002:**
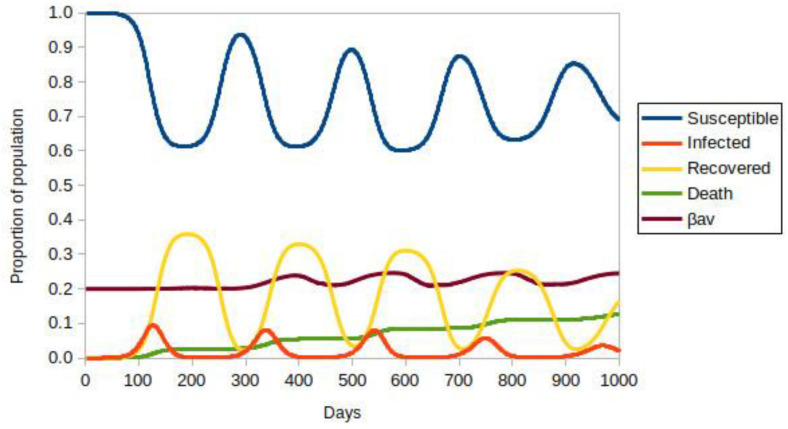
ABM-SIR dynamic of COVID-19 (i.e., Susceptible-Infected-Recovered Agent-Based model) evolution over 1000 days with a social isolation of 40%, adjusted for the initial population of ∼8.75 × 10^5^ individuals. There is no vaccination. Each color represents a population group, defined in the legend. The value of β_av_ is the average for all infected people.

**Figure 3 vaccines-10-00343-f003:**
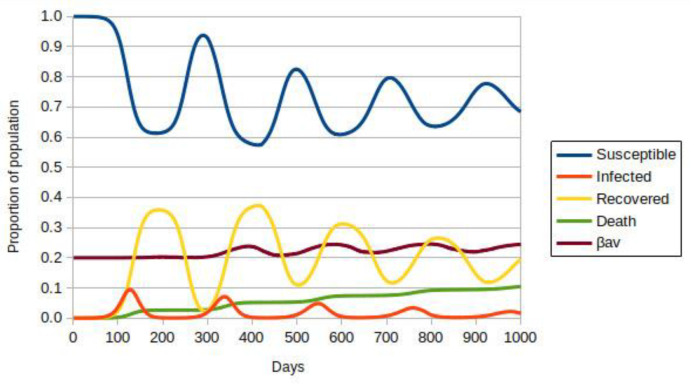
ABM-SIR dynamic of COVID-19 (i.e., Susceptible-Infected-Recovered Agent-Based model) evolution over 1000 days for social isolation of 40%, adjusted for the initial population of ∼8.75 × 10^5^ individuals. Each color represents a population group, defined in the legend. The value of β_av_ is the average for all infected people. The vaccination rate is 1/1000 of susceptible or recovered individuals per day.

**Figure 4 vaccines-10-00343-f004:**
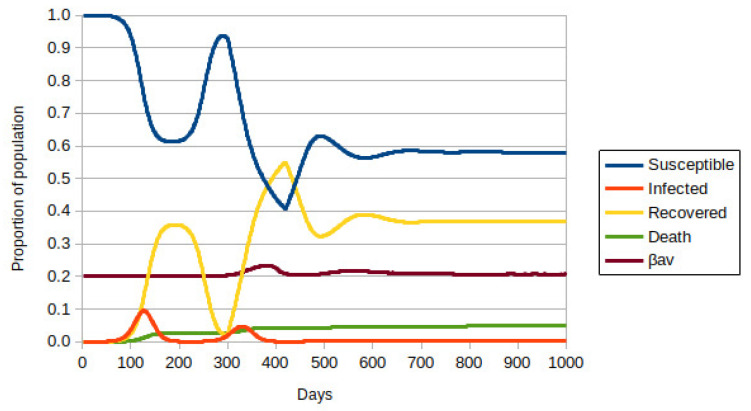
ABM-SIR dynamic of COVID-19 (i.e., Susceptible-Infected-Recovered Agent-Based model) evolution over 1000 days for social isolation of 40%, adjusted for the initial population of ∼8.75 × 10^5^ individuals. Each color represents a population group, defined in the legend. The value of β_av_ is the average for all infected people. The vaccination rate is 1/200 of susceptible or recovered individuals per day.

**Figure 5 vaccines-10-00343-f005:**
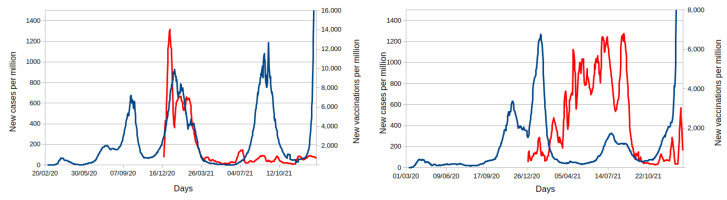
Real data for new cases (blue line) and new vaccinations (red line) for Israel (left plot) and Portugal (right plot). All curves are standardized per million and are not cumulative.

**Figure 6 vaccines-10-00343-f006:**
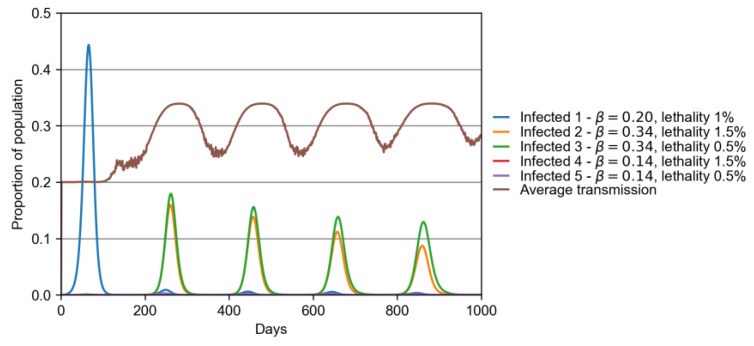
Presence of virus in population, same scenario of Appendix A. This plot show how rapidly infection caused by variants 2 and 3 overcome the original one from the second wave.

**Table 1 vaccines-10-00343-t001:** Contagion (β) and lethality of the original and variant strains used in the second model.

Variant	β	Lethality (%)
1 (original)	0.2	1.0
2	1.7 × 0.2	1.5
3	1.7 × 0.2	0.5
4	0.7 × 0.2	1.5
5	0.7 × 0.2	0.5

## Data Availability

The codes of the model are available at Appendix A.

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
