# Peer review of "Successive Pandemic Waves with Different Virulent Strains and the Effects of Vaccination for SARS-CoV-2"

_vaccines, 2022, doi:10.3390/vaccines10030343_

Round 1

Reviewer 1 Report

Castro-e-Silva et. al. presented their mathematical modeling of the effects of virus variants and vaccination rate on the spreading of the SARS-CoV-2 virus. The results presented in the manuscript have provided some theoretical understanding of the importance of a high vaccination rate and the shift in the dominant virus variants to more contagious strains. There are a few points that I suggest the authors look into.

  1. Although this is only an approximate model, the authors should consider adding some validation tests for their model against real data.
  2. The authors designed an asymmetric model where an infected person cannot infect others who have arrived earlier. This is highly unrealistic and may significantly reduce the effects of modeled infected visitors who only have one day to affect the local residents.
  3. In the model designed by the authors, an infected person is free to move. However, in practice, the infected are usually hospitalized or quarantined which prevents them from infecting others. I suggest the authors add a new version of their model accordingly.
  4. In reality, a person that gets vaccinated against some strains may also be infected by a new strain. I suggest the authors consider adding multiple vaccines into the simulation model where each vaccine is effective against a single virus strain.
  5. The authors should be consistent in terms of the x-axis label. Figures 2 and 4 used “Days” while figures 3 and 5 used “Time steps”.

Author Response

  Thank you for the opportunity to provide a better version of our manuscript and thank both reviewers for helping us in producing it so. All suggestions and criticisms were very positive and helped us to tackle some problems and improve the readability of the text. In addition, we reviewed some specific parts of the text in order to reduce redundancy and to double check language. All changes are marked. Please see the attachment.

Response to Reviewer 1 Comments

Point 1: Although this is only an approximate model, the authors should consider adding some validation tests for their model against real data.

Response 1: We are using real epidemiological data from Israel and Portugal to answer both topic (A) and question 1 from reviewer 1. Was added a text in discussion (page 12), and a new graph (Figure 6) with real data of Israel and Portugal.

Point 2: The authors designed an asymmetric model where an infected person cannot infect others who have arrived earlier. This is highly unrealistic and may significantly reduce the effects of modeled infected visitors who only have one day to affect the local residents.

Response 2: This asymmetry is counterbalanced by a continuous stream of incoming visitors with a fixed proportion of infected individuals. This stream assures the contagious introduced by visitors repeats in a daily scale. In other words, while each visitor has one day to infect local residents, this dynamic is repeated again day after day. It may be artificial but it is a simpler mathematical way to reproduce the reality of an open and uncontrolled city entrance, thus, not at all unrealistic. This rationale is explained in page XX line XX.

Point 3: In the model designed by the authors, an infected person is free to move. However, in practice, the infected are usually hospitalized or quarantined which prevents them from infecting others. I suggest the authors add a new version of their model accordingly.

Response 3: We introduced now: “Coronavirus has a high transmissibility during pre-illness period, and the model assumed the 7 days transmitting before causing any disease, which was estimated by the pandemic for previous and also delta variant. Likewise, transmission may happen from assymptomatic ones [26-28]”,  (highlight text on page 4, line XX). Hence, this model emphasizes the transmissibility by pre-symptomatic and asymptomatic, which are people with no constraint in the movement. Then, we input a symplification for sake of the transmission dynamic: after getting disease a person or recovered and does not transmit anymore, or just die. By the way, is the very mobility of pre-symptomatic and asymptomatic through the global network airports that makes Covid-19 spread to the world and became pandemic. Correlation between infection spreading and airports is a topic studied by authors and it can be viewed in references [1] and [2].

Point 4: In reality, a person that gets vaccinated against some strains may also be infected by a new strain. I suggest the authors consider adding multiple vaccines into the simulation model where each vaccine is effective against a single virus strain.

Response 4: When a simulation model is built, many considerations regarding realism must be kept aside in order to catch fundamental aspects of the aimed dynamic, but also to get feasible to write up the codes from which computer simulation is applied. Still, our present model’s approach to immunization followed what actually happened in the world. The vaccines used until now were developed using as basis the wild type, original virus first found in Wuhan, and have an lower average success over all subsequent variants. For sake of the simulation, that represents a slowing down in spreading the immunity among the population, but not a change in the overall output related to a faster than naturally acquired immunity by infection. Thus, the model focuses in two main goals: (1) how a vaccine policy (rate of immunization) can change the following waves of infection in a single strain-vaccine scenario (a simplification that only shorten up the time and simplified the distribution of immunization acquired) and (2) how different strains compete among themselves freely, in the absence of any kind of immunization policy. A discussion about vaccines policies and distribution was added at page 11, section 4.

Point 5: The authors should be consistent in terms of the x-axis label. Figures 2 and 4 used “Days” while figures 3 and 5 used “Time steps”.

Response 5: We have corrected figures 3 and 5 (now it is figure 6) x-axis to “Days”, y-axis was changed to “Proportion of population”as well.

Reviewer 2 Report

The authors developed a ABM-SIR model and discussed the effect of variant viruses and vaccination rate in the pandemic dynamics. Although I am intimately familiar with the topic of this paper, I found some of the authors' explanations difficult to follow.

Major points:

  1. Because there are so many variables (Characteristics of variants, vaccine efficacy, duration of vaccine effect etc.), it will be more useful to analyze the real-life data than simulation.
  2. How about compare the situation in Israel and similar size of countries which have low vaccination rate? Does the vaccination really change the pandemic dynamics?
  3. Please discuss method of vaccine distribution. Wide and low supply to every country (Single-immunization) or focused intensive supply (Immunize every 3 months)?

Author Response

Point 1:  Because there are so many variables (Characteristics of variants, vaccine efficacy, duration of vaccine effect etc.), it will be more useful to analyze the real-life data than simulation.

Response 1: In a research based on mathematical models, parameter-driven patterns that could help us to understand the date prevail upon the multitude of variables in the real life. In order to describe and produce reliable predictions it is needed to reduce the simulated scenario to one likely to be properly interpreted. Thus, to study intensity and duration of a host-pathogen dynamics, in controlled scenarios having virus strains with a reduced life traits of greater relevance (basically, different lethality and transmissibility), and then exposing the host population to a sudden spread of immunity (vaccination) that is able to reduce death and contagious, no other real variable is welcome than those we used. Variances in vaccine types, efficiency, duration of acquired immunity etc are reduced to an average value capable to reproduce the pandemic scenario. Also, computer models allowed us to rapidly test changes in scenarios caused by our parameters, by producing outputs used to test a specific hypothesis. Using a model within an universe of N variables makes possible to keep fixed (N-1) while it changes only 1. That approach is impossible if we deal only with real-life data. In other words, the model test the strength and actual influence of each simulated variable, preventing confounding scenarios and uncontrolled effects, typically related to the real data studies.

Point 2:  How about compare the situation in Israel and similar size of countries which have low vaccination rate? Does the vaccination really change the pandemic dynamics?

Response 2: This point have 2 questions, then we split the answers in topics (A) and (B). We now used real epidemiological data from Israel and Portugal to answer both topic (A) and question 1 from reviewer 1. Was added a text in discussion (page 12), and a new graph (Figure 6) with real data of Israel and Portugal.

Topic (B) answer: We have no doubt that vaccination completely changes the dynamics of the pandemic, even though vaccines may not protect against the most recent variants (delta and omicron) as well as they did for previous ones, they do protect against serious illness and, consequently, hospitalization and deaths. We added: “Vaccination, more than natural infection, in addition to inducing the production of neutralizing antibodies, establishes a cellular immune response by activating memory T cells, which leads to a favorable clinical outcome and considerably reduces the number of deaths. The most recent data have clearly shown that 70% of ICU admissions in  countries in the European community, the USA and even in Latin America (Brazil) are people who have either not been vaccinated or have not had the protocol of two or three complete vaccine doses. Finally, vaccination changes the dynamics as it reduces the time of infection in vaccinated individuals and consequently interferes with the dynamics of the virus in the human body, preventing serious, long-term infections and, consequently, reducing the risks of originating new variants. It is worrying the existence of countries with low vaccination coverage, as some countries in the European community and several others in Africa. Particularly for some African countries, there is still a concern about the number of HIV+ people without access to treatment, thus immune suppressed, which may become sources for emergence of new variants.” In page XXX, line XX.

Point 3:  Please discuss method of vaccine distribution. Wide and low supply to every country (Single-immunization) or focused intensive supply (Immunize every 3 months)?

Response 3: The following text was added to discussion topic, page 11. “

Our mathematical model was built considering a continuous immunization, with a single dose and with a regular immunization rate in a scenario with no vaccine shortage. In addition, our mathematical model considered that vaccination was able to generate a four-month temporary immunity with only one dose, so immunization and booster doses every six months were not included for the calculations that we propose in this work. But it is important considering that the emergence of delta and now omicron further reinforces the importance of access to COVID-19 vaccines, globally and equitably for the health of all. Constrained vaccine supply has driven opportunities for SARS-CoV-2 to mutate and be more infectious.

The emergence of omicron has emphasized that further delay in widely delivering at least first two doses is fraught with peril for all [36]. On the other hand, our model reinforces the need to plan more strategic and specific vaccination schedules and the need to review current vaccines and to design vaccines with the ability to prevent infection. The current vaccines have been overtaken by variants of concern and vaccine escape, despite maintaining protection against hospitalization and deaths. Nonetheless, we agree with the recommendation that in countries with a high prevalence of previous infections and a low proportion of over 60 years old people, to prioritize delivering the first dose will have the greatest effect on preventing severe COVID-19. Conversely, in countries with a low prevalence of previous infections and a high proportion of old population, protection against severe disease in adults requires at least two doses, likewise booster doses in people who are severely immunocompromised or older than 60 years. Evidence suggests that although booster doses for all adults might prevent severe diseases, it also could compromise timely global availability of first doses [36].

.

Round 2

Reviewer 1 Report

I am okay with the authors' revised manuscript. 

Author Response

Thank you for all questions e comments, they are all very positive and help us to improve the manuscript.  A new version of the manuscript, was uploaded. 

Reviewer 2 Report

Well revised by including real data, but there are two Figure 6 now.

Author Response

We appreciate all referee's comments and questions. The duplicate of Figure 6 was solved. There are now Figure 5 and Figure 6.